# YOLOv3-Lite: A Lightweight Crack Detection Network for Aircraft Structure Based on Depthwise Separable Convolutions

**Yadan Li** [1,2] **, Zhenqi Han** [2,*]**, Haoyu Xu** [3]**, Lizhuang Liu** [2,*]**, Xiaoqiang Li** [1] **and Keke Zhang** [4]

1   School of Computer Engineering and Science, Shanghai University, Shanghai 200444, China; liyadan@sari.ac.cn (Y.L.); xqli@shu.edu.cn (X.L.)
2   Shanghai Advanced Research, Chinese Academy of Sciences, Shanghai 201210, China
3   Lenovo Research, Shanghai Branch, Shanghai 201210, China; xuhaoyu@sina.com
4   Shanghai Engineering Center for Microsatellites, Shanghai 201210, China; liyadan5201@gmail.com
*   Correspondence: hanzq@sari.ac.cn (Z.H.); liulz@sari.ac.cn (L.L.)

**Abstract:** Due to the high proportion of aircraft faults caused by cracks in aircraft structures, crack inspection in aircraft structures has long played an important role in the aviation industry. The existing approaches, however, are time-consuming or have poor accuracy, given the complex background of aircraft structure images. In order to solve these problems, we propose the YOLOv3-Lite method, which combines depthwise separable convolution, feature pyramids, and YOLOv3. Depthwise separable convolution is employed to design the backbone network for reducing parameters and for extracting crack features effectively. Then, the feature pyramid joins together low-resolution, semantically strong features at a high-resolution for obtaining rich semantics. Finally, YOLOv3 is used for the bounding box regression. YOLOv3-Lite is a fast and accurate crack detection method, which can be used on aircraft structure such as fuselage or engine blades. The result shows that, with almost no loss of detection accuracy, the speed of YOLOv3-Lite is 50% more than that of YOLOv3. It can be concluded that YOLOv3-Lite can reach state-of-the-art performance.

**Keywords:** depthwise separable convolution; YOLOv3; feature pyramid; aircraft structure crack detection

## 1. Introduction

Identifying crack defects during the inspection of aging aircraft is of vital importance to the safety of the aircraft. The main reason for the China Airlines Flight 611 [1] air crash was the presence of cracks, which had not been completely flattened and became more and more serious.. They eventually caused the body of the planeto disintegrate in mid-air. This paper focuses on the detection of cracks in the aircraft structures, such as their engines or fuselage surface.

The method used at present for aircraft crack detection is visual inspection. This method involves a great amount of human labor, during which technicians must be fully focused in order to accurately find all the damaged areas. With the eruption in development of airlines in recent years, traditional visual inspection methods cannot fulfill the vast demand, as well as the high accuracy requirements, for crack inspection, due to the possibility of missing or false detection caused by human fatigue. Therefore, an automatic and intelligent method which can extract crack information from images or videos is necessary in order to reduce human labor and to help servicing engineers to speed up the inspection process and improve accuracy simultaneously.



Therefore, the study of aircraft crack detection has attracted the interest of many researchers. These works will be illustrated in two aspects. One is the traditional detection method, which depends on special hardware facilities, and the other is the visual detection method, based on deep learning.

The first kind of crack detection methods depends on specialized hardware devices. A resonant ultrasound spectroscopy apparatus was provided for detecting crack-like flaws in components in [2]. Searle et al. applied an aircraft structural health-monitoring system to detect damaged areas in a full-scale aircraft fatigue tests [3]. Kadam et al. [4] used a self-diagnosis technique to detect the cracking and de-bonding of the permanently embedded lead zirconate titanate (PZT). Cracks are reported as key in [5], which use the first-order reliability method (FORM + Fracture) to alleviate the computational cost of probabilistic defect-propagation analysis [6–10], which is commonly used to detect cracks using a special hardware device.

Some algorithms for crack detection are based on deep learning. Recently, deep learning has achieved great improvements in various visual tasks such as object classification [11–13], and object detection [14–16]. Deep learning fundamentally changes the ways to tackle some traditionally hard or intractable visual tasks and has produced many successful applications. Therefore, the application of deep learning in crack detection has emerged. Recent studies have designed deep learning-based crack recognition methods: [17] proposed a 5-layers Convolutional Neural Network (CNN) to detect cracks, while [18] evaluated five CNN architectures for corrosion detection from input images, in all of which large input images were cropped into small images of a fixed size. The CNNs were applied to classify whether cracks or corrosion were contained in each small input image. Ref. [19] proposed an aircraft engine borescope crack detection and segmentation system based on deep learning. Ref. [20] proposed an algorithm which learned hierarchical convolutional features for crack detection, which was actually an edge detection method. Although the crack features were extracted well, their detection scenarios are very simple, so the method could not be applied to the complex scene detection of aircraft structural cracks.

All of the crack detection methods mentioned above did not meet the needs of assistant technicians in order to complete rapid and accurate crack detection in aircraft structures. First, because the crack detection of aircraft structure is typically conducted in an outdoor environment, the detection equipment must be portable. However, the methods mentioned above require specialized professional and complex equipment for crack detection, which is inconvenient in aircraft inspection. Second, aircraft structures, especially the internal background of the engine, are very complex, and the performance of the algorithm in complex environments is very demanding. However, the background of the crack images used in the methods mentioned above is very clean, as is shown in Figure 1. Most of the cracks in other works are similar to those shown in the last two figures. Third, crack detection in aircraft structures needs to be completed accurately in a limited time. So, the offline detection speed of the algorithm needs to be close to real-time and guarantee high accuracy as well. However, when the efficiency of the above methods was improved, the accuracy decreased significantly. These two indicators cannot be taken into account simultaneously by the above methods.

To solve the problems mentioned above, we propose an aircraft structure crack detection algorithm: YOLOv3-Lite. This method is a deep learning-based framework, which can accurately and efficiently detect crack damage on an aircraft structure. Compared with the previous crack detection method, YOLOv3-Lite shows that it can reach state-of-the-art performance. It can run on a mobile device due to its light-weight characteristics. On the premise of guaranteeing a certain accuracy rate, the speed has been greatly increased. In addition, the speed improvement of YOLOv3-Lite refers to the shortening of offline detection time.

The algorithm pipeline can be described by three parts. Firstly, set is divided into the training set, the validation set, and the test set. The data is processed in a form that can be received by YOLOv3-Lite. Secondly, the YOLOv3-Lite architecture is built, which adopt depthwise separable convolution, feature pyramid, and YOLOv3 [21] methods. Additionally, transfer learning is introduced to reduce the required amount of data and yield high accuracy. We evaluate the model by comparing with baseline

performance and achieve comparable AP scores and competitive efficiency as well. Finally, the model is applied to test data. The pipeline is shown in Figure 2.

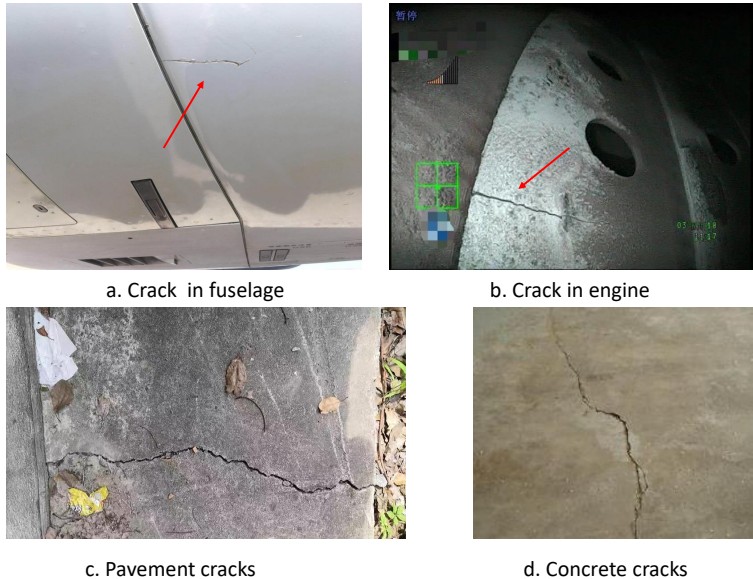

a. Crack in fuselage      b. Crack in engine

c. Pavement cracks      d. Concrete cracks

**Figure 1.** A comparison of aircraft structure cracks and crack images used in other works. (**a**) crack in fuselage, (**b**) crack in engine, (**c**) cracks in the pavement, (**d**) cracks in concrete.

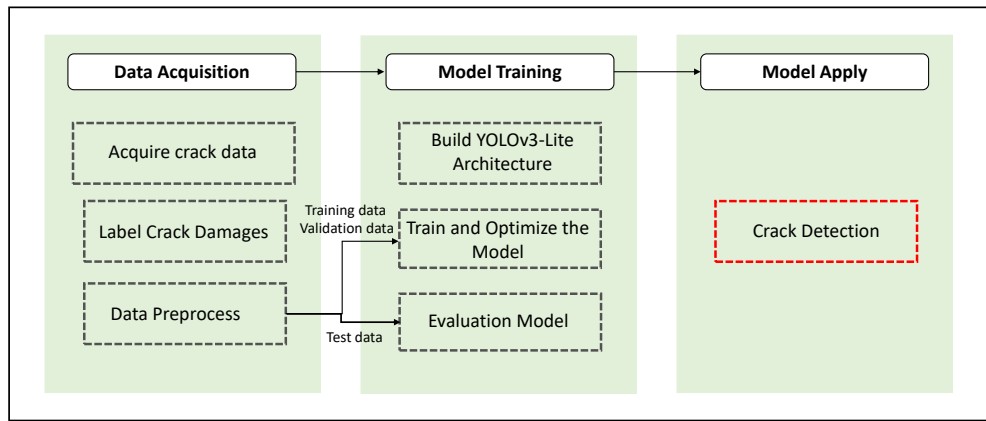

**Figure 2.** The overall architecture of the proposed YOLOv3-Lite algorithm.

The special contribution of this paper are as follows:

1. The proposed algorithm is a lightweight network with fewer parameters, such that it can be migrated to mobile devices.
2. It is a novel aircraft structure crack detection network which can detect the different parts of an aircraft and can detect different types of an aircraft, which means it has good generalization performance.
3. The crack detection network is combined with depthwise separable convolution and feature pyramids so it is fast and accurate.

The rest of the paper is organized as follows: Section 2 explains the YOLOv3-Lite algorithm. Section 3 shows some experiments using YOLOv3-Lite and discusses the experimental results. Finally, in Section 4, some future works are presented and the paper is concluded.

## 2. Aircraft Structural Crack Detection Method

In this section, we start with depthwise separable convolution, which is used by YOLOv3-Lite for a backbone network. Then, YOLOv3 is introduced, which is adopted for bounding box regression. Finally, the architecture of the aircraft structural crack detection network is proposed.

### 2.1. Depthwise Separable Convolution

Deep separable convolution is a convolution method proposed by MobileNet [22], which can greatly reduce the parameters and achieve the same effects as standard convolution. Depthwise separable convolutions are a form of factorized convolutions. They factorize a standard convolution into a depthwise convolution and a $1 \times 1$ convolution called a pointwise convolution. In depthwise convolution applies a single filter to each input channel and, then, a $1 \times 1$ convolution is applied to combine the outputs of the depthwise convolution by the pointwise convolution: compare this to a standard convolution whose input is convoluted and combined into a new set of outputs in one step the depthwise separable convolution splits this process into two layers, a separate layer for filtering and a separate layer for combining. This factorization has the effect of drastically reducing computation time and model size. The convolution principle of depthwise separable convolution and the standard convolution is shown in Figure 3.

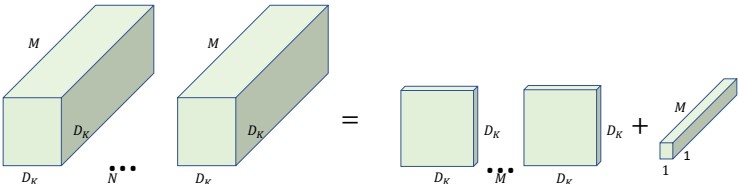

**Figure 3.** Standard convolution kernels: $N \times D_K \times D_K \times M$ (**left side**) and $M \times D_K \times D_K$ depthwise convolution kernels and $1 \times 1 \times M$ pointwise convolution kernels (**right side**).

A standard convolution layers takes a feature map of $D_F \times D_F \times M$ as input, where $D_F$ is the spatial width and height of a square feature map and $M$ is the number of input channels, which is parameterized by a convolution kernel $K$ of size $D_K \times D_K \times M \times N$ where $D_K$ is the spatial dimension of the kernel, $M$ is the number of input channels, and $N$ is the number of output channels. The output feature map for standard convolution, assuming a stride of one and padding computed as: $G_N = \sum K_{(N,M)} \times F_M$. Thus, the computation cost of the standard convolution is $D_K \cdot D_K \cdot M \cdot N \cdot D_F \cdot D_F$.

According to this formula, we can draw the conclusion that the number of input and output channels, the kernel size, and the feature map size contribute a great deal to the computational cost. However, depthwise separable convolutions break the interaction between the number of output channels and the size of the kernel. It is made up of two layers which are depthwise convolutions $d\_w$ and pointwise convolutions $p\_w$. A single filter is applied to each input channel depthwise and then by pointwise convolution, a simple $1 \times 1$ convolution, which is conducted to create a linear combination of the output of the depthwise layer. The output feature map for depthwise convolution is computed as:

$$\hat{G}_M = \sum K_{(1,M)} \times F_M. \tag{1}$$

The computation cost of the depthwise separable convolution is $D_K \cdot D_K \cdot M \cdot D_F \cdot D_F$.

By decomposing the convolution process into two steps, the computational complexity is reduced by:

$$\frac{D_K \cdot D_K \cdot M \cdot D_F \cdot D_F + M \cdot N \cdot D_F \cdot D_F}{D_K \cdot D_K \cdot M \cdot N \cdot D_F \cdot D_F} = \frac{1}{N} + \frac{1}{D_K^2}. \tag{2}$$

Therefore, the computational effort of $3 \times 3$ depthwise separable convolutions will be reduced by 8 to 9 times, compared to standard convolution.

### 2.2. YOLOv3

YOLOv3 is a deep convolutional architecture designed for object detection, which uses the darknet as the feature extract network and then using dimension clusters as anchor boxes for predicting bounding boxes of the system. Four co-ordinate $(t_x, t_y, t_w, t_h)$ values for each bounding box and their confidence scores are output from the input image directly through the regression operation, as well as class probabilities. Confidence scores represent the precision of the predicted bounding box when the grid contains an object.

At the training stage, the three feature maps $(13 \times 13, 26 \times 26, 52 \times 52)$ output from the feature extract network. Taking the feature map of size $13 \times 13$ as an example, the proposed method divides the feature map into $13 \times 13$ grids. Each grid takes charge of the object detection in case the ground truth is contained in it. the predictions will be obtained as:

$$\begin{aligned} b_x &= \sigma(t_x) + C_x \\ b_y &= \sigma(t_y) + C_y \\ b_w &= p_w e^{t_w} \\ b_h &= p_h e^{t_h}, \end{aligned} \tag{3}$$

where the $(C_x, C_y)$ denote that the center of an object is detected in a grid which is offset from the top left corner of the feature map; $(p_w, p_h)$ denotes the width and height of the anchor box prior, respectively; and $(t_x, t_y, t_w, t_h)$ are the four offset co-ordinate predicted by the network. Using a sigmoid to compress $t_x$ and $t_y$ to $[0, 1]$, the target center can be effectively ensured to be in the grid cell executing prediction. A bounding box, with dimension priors and location prediction, is shown in Figure 4.

The loss function of YOLOv3 is composed of three parts: a co-ordinate prediction error (terms 1 and 2), a confidence score (terms 3 and 4) which is the intersection over union (IoU) error, and a classification error (the last term). This loss is defined as follows:

$$\begin{aligned} Loss = &\lambda_{coord} \sum_{i=0}^{S^2} \sum_{j=0}^{B} 1_{ij}^{obj} [l(x_i, \hat{x}_i) + l(y_i, \hat{y}_i)] \\ &+ \lambda_{coord} \sum_{i=0}^{S^2} \sum_{j=0}^{B} 1_{ij}^{obj} [(\sqrt{w_i} - \sqrt{\hat{w}_i})^2 + (\sqrt{h_i} - \sqrt{\hat{h}_i})^2] \\ &+ \sum_{i=0}^{S^2} \sum_{j=0}^{B} 1_{ij}^{obj} l(C_i, \hat{C}_i) \\ &+ \lambda_{noobj} \sum_{i=0}^{S^2} \sum_{j=0}^{B} 1_{ij}^{noobj} l(C_i, \hat{C}_i) \\ &+ \sum_{i=0}^{S^2} 1_{i}^{obj} \sum_{c \in classes} l(p_i(c), \hat{p_i(c)}), \end{aligned} \tag{4}$$

where $1_{ij}^{obj}$ indicates that the target is detected by the $j_{th}$ bounding box of grid $i$. In order to increase the loss from bounding box coordinate predictions and decrease the loss for confidence predictions for boxes that do not contain objects, the parameters $\lambda_{coord}$ and $\lambda_{noobj}$ are introduced and both set to 5. Then, $\hat{x}_i, \hat{y}_i, \hat{w}_i, \hat{h}_i$ are the predicted bounding box parameters of center co-ordinates and box size. $x_i, y_i, w_i, h_i$ are the actual parameters, $\hat{C}_i$ is the prediction of the confidence score, $C_i$ is the true data; $p_i(c)$ indicates the true value of the probability of the object in grid $i$ belonging to class $C$; and $\hat{p}_i(c)$ is

the predicted value. Except for the box size error, which uses the mean square error, the others use the binary cross-entropy loss $l(a, \hat{a})$ which is defined as follows:

$$l(a, \hat{a}) = -a_i log\hat{a}_i + (1 - a_i)log(1 - \hat{a}_i). \tag{5}$$

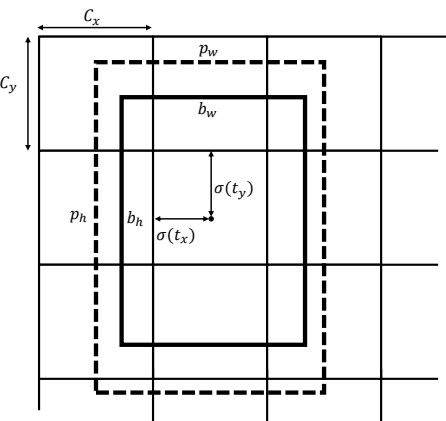

**Figure 4.** YOLOv3 predicts the width and height of the box as offsets from cluster centroids and predicts the center coordinates of the box relative to the location of filter application using a sigmoid function.

### 2.3. YOLOv3-Lite

In YOLOv3-Lite, the backbone network is designed, inspired by depthwise separable convolution. As the crack sizes vary greatly (i.e., there are centimeter-scale and decimeter-scale cracks), we extract features from three scales, using a similar concept to the feature pyramid network [23]. In order to realize the fusion of low-resolution and semantically strong features with more effective high-resolution features, we concatenate the feature pyramids. From our base feature extractor, we add one convolutional layer, which can improve the effectiveness of the network, as determined experimentally. The YOLOv3-Lite can greatly reduce the number of parameters and also achieve higher accuracy in portable equipment. The overall architecture of the network is shown in Figure 5.

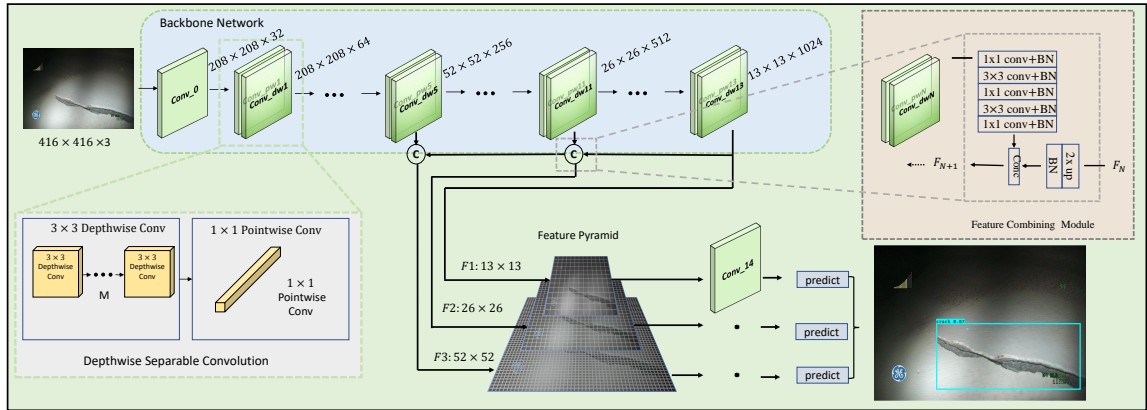

**Figure 5.** Network structure diagram of the YOLOv3-Lite model. The input is $416 \times 416 \times 3$. Crack features are extracted by the backbone network, designed using depthwise separable convolution. The feature pyramid is calculated for detecting cracks at different scales. Finally, the network outputs the detection results.

2.3.1. Backbone Network

Considering the need to migrate the algorithm to mobile devices, the backbone of the detection network is built on depthwise separable convolution to compose the basic convolution unit. The whole feature extraction network structure is defined in Table 1; there are 13 depthwise separable convolutional layers, where each layer contains one depthwise convolution and one pointwise convolution layer. All layers are followed by a batchnorm [24] and ReLU non-linearity layer. Downsampling is handled with stride convolution in the depthwise convolutions, as well as in the first layer.

**Table 1.** Backbone Architecture.

| Type/Stride | Filter Shape | Output Size |
|:---:|:---:|:---:|
| Conv/s2 | $3 \times 3 \times 3 \times 32$ | $208 \times 208 \times 32$ |
| Conv_dw/s1 | $3 \times 3 \times 32$ | $208 \times 208 \times 32$ |
| Conv_dw/s1 | $1 \times 1 \times 32 \times 64$ | $208 \times 208 \times 64$ |
| Conv_dw/s2 | $3 \times 3 \times 64$ | $104 \times 104 \times 64$ |
| Conv_dw/s1 | $1 \times 1 \times 64 \times 128$ | $104 \times 104 \times 128$ |
| Conv_dw/s1 | $3 \times 3 \times 128$ | $104 \times 104 \times 128$ |
| Conv_dw/s1 | $1 \times 1 \times 128 \times 128$ | $104 \times 104 \times 128$ |
| Conv_dw/s2 | $3 \times 3 \times 128$ | $52 \times 52 \times 128$ |
| Conv_dw/s1 | $1 \times 1 \times 128 \times 256$ | $52 \times 52 \times 256$ |
| Conv_dw/s1 | $3 \times 3 \times 256$ | $52 \times 52 \times 256$ |
| Conv_dw/s1 | $1 \times 1 \times 256 \times 256$ | $52 \times 52 \times 256$ |
| Conv_dw/s2 | $3 \times 3 \times 256$ | $26 \times 26 \times 256$ |
| Conv_dw/s1 | $1 \times 1 \times 256 \times 512$ | $26 \times 26 \times 512$ |
| $5\times$   Conv_dw/s1 | $3 \times 3 \times 512$ | $26 \times 26 \times 512$ |
| Conv_dw/s1 | $1 \times 1 \times 512 \times 512$ | $26 \times 26 \times 512$ |
| Conv_dw/s2 | $3 \times 3 \times 512$ | $13 \times 13 \times 512$ |
| Conv_dw/s1 | $1 \times 1 \times 512 \times 1024$ | $13 \times 13 \times 1024$ |
| Conv_dw/s1 | $3 \times 3 \times 1024$ | $13 \times 13 \times 1024$ |
| Conv_dw/s1 | $1 \times 1 \times 1024 \times 1024$ | $13 \times 13 \times 1024$ |

We designed the backbone network that generates three-layer feature pyramid which contains three scales of feature maps in order to detect cracks of different sizes. The size of the network input image is $416 \times 416$, to which the multilayer depthwise separable convolution is applied. The last layer of the network outputs the $13 \times 13$ feature map, which is marked as $f1$. Layer 11 concatenates with $f1$ after up-sampling, then outputs the $26 \times 26$ feature map, which is marked as $f2$. Finally, the $52 \times 52$ feature map is computed by layer 5, concatenated with $f2$ after up-sampling. The concatenated parts are built by the residual network, which can combine low-level information and high-level Semantic Information. In this connection mode, the network can learn the crack features effectively. Finally, the pyramid feature map contains these three layers. Due to the large receptive field of the $13 \times 13$ feature map, it can detect large-sized cracks. On the contrary, the $52 \times 52$ feature map has smaller receptive field, such that small-sized crack can be detected by this layer.

2.3.2. Bounding Box Prediction

After extracting the features of the cracks, we regress the bounding box for crack detection, based on YOLOv3. There are nine appropriate prior anchor boxes which are clustered with training data labels by a K-means clustering algorithms. Each of the three feature maps has three anchor boxes. As multilayer convolution layers can reduce the resolution of an image, the last layer of the network has the lowest resolution. Thus, small crack features may be lost, while the large crack features still

exist. Therefore, the last layer of the network is more sensitive to large cracks. At the same time, its receptive field is also the largest, so three larger anchor boxes are allocated to the 13 × 13 feature map obtained by the last layers for detecting large cracks. Three medium-size anchor boxes are for the 26 × 26 feature map and the smaller anchor boxes are for the 52 × 52 feature map, for detecting small cracks. Four co-ordinate values for each bounding box and their confidence scores output by YOLOv3. As we only have a crack class, the class probability is 1. Confidence scores represent the precision of the predicting bounding box when the grid contains the crack. Only the anchor box which has the highest confidence scores will be selected for regression; the other two anchors will be adopted by the non-maximum suppression algorithm. Then, the selected anchor box will gradually regress to the position and size of ground truth at the training stage.

## 3. Experimental Results and Analyses

Our experiments are aimed at crack detection in aircraft structures, including in the surface of the fuselage, engine blades, and in other parts. In this section, firstly, we will describe the data set composition and some training settings. Secondly, the effect of YOLOv3-Lite is demonstrated by several experimental results. Thirdly, the detection performance among YOLOv3-Lite, MobileNet-SSD [25], YOLO-Tiny and YOLOv3 will be compared.

### 3.1. Dataset Composition and Characteristic

Our dataset consists of two parts, one of which is derived from aircraft structures, such as fuselage, wing, aircraft tail, and engine interior. The length of cracks ranges from 1 to more than 10 cm. About 580 images with cracks were collected by ourselves and servicing engineers which contains different cases of structural cracks in real aircraft from our partner aviation companies. The other is from the data of industrial equipment with similar cracks, as shown in Figure 6. The dataset contains 960 pictures in total. We use 800 samples as a training set and use 80 samples as a validation set for selecting a well-generalized performance model. Another 80 samples independent of the training and validation set are used as a test set.

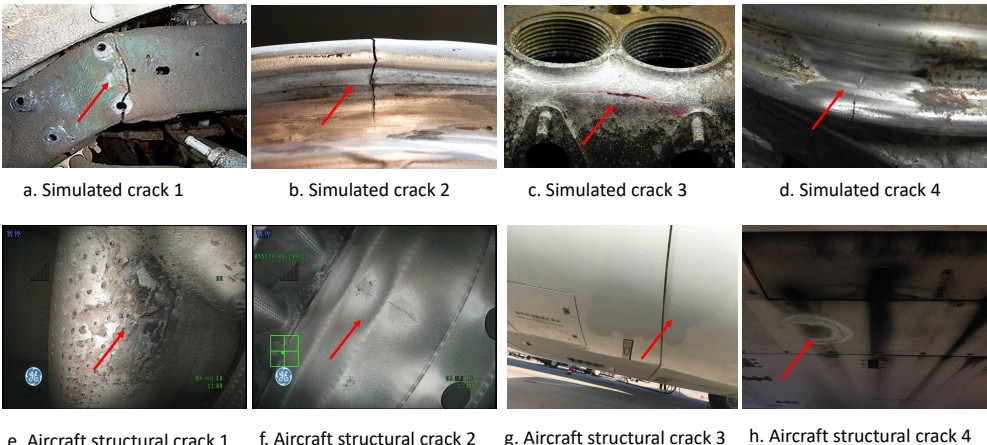

**Figure 6.** Different industrial equipments with cracks which have background interferences (**a**–**d**), aircraft structures with cracks (**e**–**h**).

As shown in Figure 6, the cracks on the fuselage and wings of an aircraft are small and light in color, so the difference between the cracks and the background is small. Meanwhile, strong illumination also increases the difficulty of crack detection in these areas. It is more difficult to detect cracks because of the complex internal structure and background of the aircraft engine.

### 3.2. Training Methodology

We implement YOLOv3-Lite using the Tensorflow 1.7.0 [26] and Python 3.7 running on an NVIDIA Tesla K20 GPU in the programming environment Linux 16.04. In YOLOv3-Lite, batch normalization is used after each convolutional layer, in both the deep separable convolution and in several standard convolution layers, which is used to speed up convergence in the training process. Firstly, we use the mechanism of transfer learning to train the network. For the backbone network, we adopt the parameters pre-trained in ImageNet. In fine-tuning, the initial learning rate is set to $1e - 3$ and is divided by 10 after every epoch. The Adam [27] is employed to update the network parameters, with a batch size of 10 in each iteration. We train the network with 300 epochs in total. Secondly, all layers are unfrozen to train in detail for 50 epochs with a batch size of 4. The initial learning rate is set to $1e - 4$ and the decay rate is the same as the fine-tuning stage. Adam is also used in this phase. The initialization parameters of the proposed model are shown in Tables 2 and 3. Using a pre-training weight training network can greatly reduce training time and experimental resources and can converge faster. Then we can adjust the weights of the whole network, through training,to make the network model more suitable in the context of aircraft structure crack detection.

**Table 2.** The initial parameters of YOLOv3-Lite in stage 1 of training.

| Size of Input Images | Batch Size | Initial Learning Rate | Decay | Training Steps |
|:---:|:---:|:---:|:---:|:---:|
| $416 \times 416$ | 10 | 0.001 | 0.1 | 300 |

**Table 3.** The initial parameters of YOLOv3-Lite in stage 2 of training.

| Size of Input Images | Batch Size | Initial Learning Rate | Decay | Training Steps |
|:---:|:---:|:---:|:---:|:---:|
| $416 \times 416$ | 4 | 0.0001 | 0.1 | 50 |

### 3.3. Evaluation Metrics

For each image, the intersection over union (IoU) between the bounding box of the detected crack and ground truth can be calculated as: $I_{oU} = \frac{A_o}{A_u}$, where $I_{oU}$ is the intersection over union, $A_o$ is the area of overlap, and $A_u$ is the area of union.

When the IoU of the predicted bounding box and ground truth is greater than a certain threshold value (e.g., 0.5), it is considered to be a true positive; otherwise, it is a false positive. A false negative is obtained by missing a crack. Then we can calculate the precision and recall. Finally, average precision (AP) which is equivalent to the area under the precision-recall curve [28] can be computed. In addition, in order to measure the efficiency of the algorithm, the offline detection time for one picture is also an important evaluation index. We compare the detection time (i.e., the offline time) and analyze the accuracy performance of several models.

### 3.4. Detection Performance of YOLOv3-Lite

Depthwise separable convolution is used to achieve the goal of the lightweight network. At the same time, the feature pyramid method is used to detect crack defects of different sizes. In order for each layer of the feature pyramid to fuse the information of the high-level and low-level layers, the feature pyramids are concatenated. The experiments show that YOLOv3-Lite can achieve high accuracy and greatly improve the detection speed. The input size of the image for our aircraft structural crack detection network is adjusted to $416 \times 416$ pixels, in order to increase the model performance. The image is detected by YOLOv3-Lite in 0.1s with an AP of 38.7%. Finally, the network outputs the detection result. Our method can quickly and accurately detect cracks in various parts of the aircraft, such as the surface of the aircraft fuselage, engine blades, aircraft tires, wing tail, and other parts. The detection results are shown in Figure 7. Whether it is obvious large-sized cracks or

small-sized cracks which are difficult to find with a nosisy background, they can be precisely detected by YOLOv3-Lite.

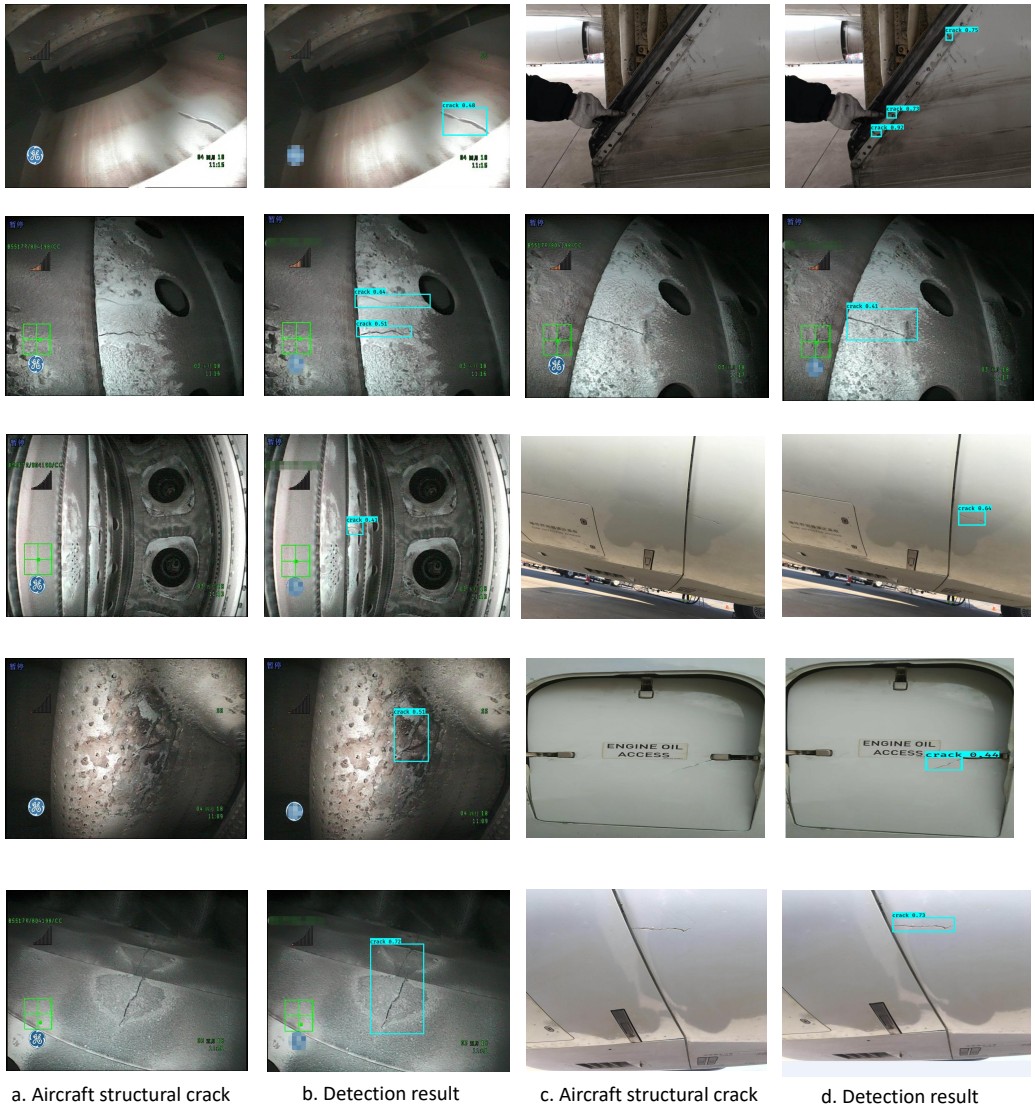

|  |  |  |  |
|---|---|---|---|
| a. Aircraft structural crack | b. Detection result | c. Aircraft structural crack | d. Detection result |

**Figure 7.** The experimental results of YOLOv3-Lite.

### 3.5. Comparison of YOLOv3-Lite with Three Modern Methods

To better verify the superiority of the improved method, we compare YOLOv3-Lite with YOLOv3, YOLO-Tiny and MobileNet-SSD. These models are evaluated in terms of detection of average precision (AP) and offline detection time. The results of the contraction experiment are shown in Table 4. The parameter quantity of YOLOv3-Lite is about 31 million, YOLOv3 is twice that of ours, at about 61 million, and YOLO-Tiny is minimum. The number of parameters of SSD-mobilenet is roughly the same as ours. However, the parameters of YOLOv3-Lite are small, the detection speed is faster than most of the algorithms, and its average precision is the highest. We adopt deep separable convolution, which reduces the number of parameters by 8–9 times, on average, with an effect similar to that of standard convolution. Therefore, the detection speed of YOLOv3-Lite is 50% faster than that of YOLOv3. As YOLO-Tiny only uses 13 standard convolution layers, its feature extraction ability is poor and AP is very low. Although we have fewer network layers, the proposed method is still much better than MobileNet-SSD as we use the way of connecting feature maps at different levels, each layer of

feature pyramid containing not only high-level semantic information, but also low-level information. Therefore, YOLOv3-Lite not only has high detection accuracy but also achieves the goal of being lightweight, which greatly improves the network detection speed.

**Table 4.** Average precision, offline detection time and the number of parameters of each network.

|  | AP | Time (s) | The Number of Parameters (Million) |
| --- | --- | --- | --- |
| YOLOv3-Lite | 38.7% | 0.125 | 31 |
| SSD-Mobilenet | 17.1% | 0.128 | 31 |
| YOLOv3 | 43.1% | 0.225 | 61 |
| YOLO-Tiny | 2.5% | 0.09 | 9 |

Although the detection time of YOLO-Tiny is the shortest, its accuracy is too low to detect cracks at all. The accuracy of YOLOv3 is the highest, but its detection speed is 50% lower than YOLOv3-Lite, so it is not a good choice. We compare the detection results of YOLOv3-Lite and SSD-Mobilenet, which are shown in Figure 8. The bounding boxes of YOLOv3-Lite can accurately frame the location of cracks, while SSD-Mobilenet not only shows a large error in the location of the boxes, but also has a very low confidence level. In addition, there are many cracks that can not be detected by the SSD-Mobilenet method. Therefore, YOLOv3-Lite performs best in these modern methods.

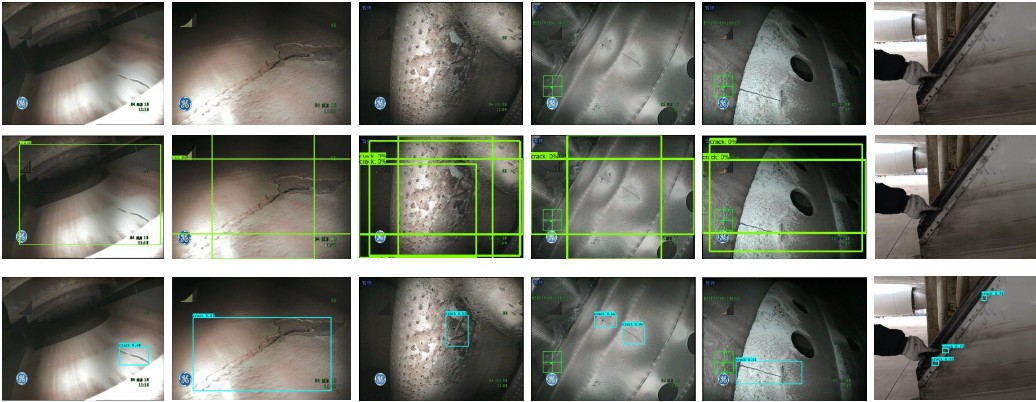

**Figure 8.** The first line is pictures of the original aircraft structures with cracks. The second line is the detection results of SSD-Mobilenet. The last line is the detection results of YOLOv3-Lite.

## 4. Conclusions

Every year, there are aircraft faults caused by crack defects. Thus, the research on crack inspection in aircraft structures is of far-reaching significance. However, The current related methods either depend on specialized hardware devices or adopt edge detection methods, which can not effectively resist the influence of background noise. We proposed the YOLOv3-Lite method to address these problems:

- We use deep separable convolution to design a feature extraction network. Using depthwise convolution and $1 \times 1$ pointwise convolution, instead of standard convolution, reduced lots of parameters.
- We adopt the idea of a feature pyramid network which combines low- and high-resolution information. This feature pyramid has rich semantic information at all levels and can be built quickly from a single input image scale.
- We use YOLOv3 for bounding box regression. The results show that the offline detection speed of YOLOv3-Lite is 50% faster than YOLOv3, and the detection accuracy and speed are better than SSD-MobileNet and YOLO-Tiny.

Therefore, YOLOv3-Lite is a light-weight, fast (50% faster), and accurate (38.7%) crack detection network, which shows that it can reach state-of-the-art performance. In order to extend YOLOv3-Lite to a general algorithm, our future work will aim optimize YOLOv3-Lite and apply it to more scenarios.

**Author Contributions:** Project administration, Y.L.; Validation, Y.L.; investigation, Y.L., Z.H. and X.L.; resources, Y.L., K.Z., L.L. and H.X.; visualization, Y.L.

**Funding:** This research was funded by Science and Technology Commission of Shanghai Municipality grant number 17511106400,17511106402 and Aerospace System Department grant number 30508020301.

**Acknowledgments:** Thanks for part of data collecting and labeling works by Chong Wang of our research group.

**Conflicts of Interest:** The authors declare no conflict of interest. The founding sponsors had no role in the design of the study; in the collection, analysis, or interpretation of data; in the writing of the manuscript, and in the decision to publish the results.

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
