# Peer review of "YOLOv3-Lite: A Lightweight Crack Detection Network for Aircraft Structure Based on Depthwise Separable Convolutions"

_applsci, doi:10.3390/app9183781_

Round 1

Reviewer 1 Report

Major comments

1. The objectives of the current research must be clearly defined. The proposed artificial neural network can detect structure cracks that can be simply recognized by visual inspection of the structure. Why do the Authors want to use a complicated artificial neural network to detect the cracks that are clearly seen? All example cracks shown in Figs. 1, 5, 6 can be detected by watching at the samples. How in practical applications the servicing engineer can benefit by using the method proposed by the Authors? After all he/she can see the cracks without the method proposed by the Authors. It would be better if the Authors develop a method that can detect such cracks that cannot be detected by other methods.

2. Implementation details of the YOLOv3-Lite algorithm developed by the Authors should be given - what software tools/programming environments do the Authors use to implement their algorithm? Have they modified existing software packages or have they developed their own code for the whole YOLOv3-Lite algorithm?

3. To properly train any artificial neural network a large database of template images is required. Often it is a problem to collect such a database. In section 3.1 the Authors explain they used a dataset consisting of 960 template images. It should be explained how have the Authors obtained these images - have they inspected 960 different cases of structure cracks in real aircrafts? Do they cooperate with any aviation companies that deal with aircraft structure cracks?

4. There are lots of language mistakes (examples are given below). The paper cannot be accepted without a thorough language revision.

Minor comments

Page 3, line 67: “It is a fast and accurate detection network...” - The Authors have not proven the speed and accuracy of their YOLOv3-Lite method yet, so it is too early to talk about this advantage of the proposed method in this place of the paper.

Page 5, Eq. (3): The symbols used in Eq. (3), e.g. $\delta(t_x)$, $c_x$ must be explained.

Page 5, Eq. (4): It seems Eq. (4) is the right side of an equation. Therefore the equation sign and the left side of this equation must be given

Page 7, lines 149-152: “Three larger anchor boxes are allocated to 13 x 13 feature map... three medium-size anchor boxes are for 26 x 26 feature map and the smaller anchor boxes are for 52 x 52 for detecting small cracks.” - It seems larger anchor boxes are allocated to 52 x 52 feature map for detecting larger cracks?

Page 7, line 162: “Finally, the effect of network detection is better than that of the existing network...” - What “existing network” do you mean?

Page 8, Eq. (6): Proper one letter notations with eventual sub- or super-indexes should be used in the mathematical formula in Eq. (6), i.e. Eq. (6) should be noted as: “$I_{oU}=\frac{A_o}{A_u}$” with the following explanation below: “where $I_{oU}$ is the intersection over union, $A_o$ is the area of overlap and $A_u$ is the area of union.”

Language mistakes

Page 1, line 3: “... is time-consuming or have poor accurate...” -> “... are time-consuming or have poor accuracy...”

Page 1, line 4: “... proposed YOLOv3-Lite, which combine...” -> “... proposed YOLOv3-Lite, which combines...”

Page 1, line 7: “The feature pyramid join together...” -> “Then the feature pyramid joins together...”

Page 1, lines 8, 9: “... YOLOv3-Lite is a fast and accurate... for aircraft structure...” -> “... YOLOv3-Lite is the fast and accurate... for an aircraft structure...”

Page 1, lines 16-18: “The air crash the reason why happened to China Airlines Flight 611... more and more serious.” - The meaning of this sentence is unclear. A proper literature reference to the China Airlines Flight 611 crash accident should be given.

Page 1, line 19: “Eventually caused...” -> “They caused...”

Page 1, line 21: “Therefore, research on aircraft...” -> “Research on aircraft...” - I would suggest to move this sentence to the beginning of section “1. Introduction”

Page 1, line 22: “... detection method depend on special...” -> “... detection method, which depends on special...”

Page 1, line 24: “Some crack detection methods are depending...” -> “Some crack detection methods depend...”

Page 1, line 24: “Resonant ultrasound spectroscopy apparatus...” -> “A resonant ultrasound spectroscopy apparatus...”

Page 1, line 25: “Jan Serle et al. applied aircraft...” -> “Serle et al. applied an aircraft...”

Page 1, line 26: “Kadam et al. A self-diagnosis technique... is used to detect... are commonly used to detect crack by a special hardware device” -> “Kadam et al. used a self-diagnosis technique... to detect... are commonly used to detect crack by a special hardware device” - The meaning of this sentence is unclear.

Page 1, line 31: “... in a various visual task such as...” -> “... in various visual tasks such as...”

Page 2, line 34: “Some researches design deep... methods recently.” -> “Some researchers have designed deep... methods recently.”

Page 2, line 35: “... a 5 layers CNN to detect cracks...” - The acronym “CNN” should be explained.

Page 2, line 35: “... for detection corrosion...” -> “... for corrosion detection...”

Page 2, line 39: “... an algorithm that learning hierarchical...” -> “... an algorithm that learns hierarchical...”

Page 2, line 40: “... edge detection method although the crack feature...” -> “... edge detection method. Although the crack feature...”

Page 2, lines 46, 47: “... interference factor affecting aircraft structure detection.” -> “... interference factors affecting aircraft structure crack detection.”

Page 2, line 49: “... the crack image of other works.” -> “... the crack image of other structures.” - What do you mean by “works”?

Page 2, caption of Fig. 1: Some parts of the caption of Fig. 1 can be moved to the text above Fig. 1:

“Figure 1. The comparable of aircraft... and the crack images of other works. There are a lot of noise... are only cracks. (a) is a crack in fuselage. (b) is a crack in engine. (c) is cracks on the pavement. (d) is cracks on concrete. Most of the cracks in other works... last two figures” -> “The comparison of aircraft... and crack images of other structures: (a) crack in fuselage, (b) crack in engine, (c) cracks in the pavement, (d) cracks in concrete”

“There are a lot of noise in the airplane structure.” - Move this sentence to the text above Fig. 1.

“Most of the cracks in other works are similar to those shown in the last two figures.” -> “Most of the cracks in other structures are similar to those shown in the last two figures.” - Move this sentence to the text above Fig. 1.

Page 2, line 50: “... proposed a crack detection of aircraft structure algorithm...” -> “... proposed an aircraft structure crack detection algorithm...”

Page 2, line 52: “So, compared with the previous crack detection method...” -> “Compared with the previous crack detection method...” - Which previous crack detection method do you mean in this sentence?

Page 3, line 55: “... described from three parts. Firstly, we acquire crack data... damage then the data” -> “... described by three parts. Firstly, crack data is acquired... damage. Then the data...”

Page 3, line 56: “... into the training set, validation set, and test set.” -> “... into the training set, the validation set, and the test set.”

Page 3, line 57: “... is processed as a form...” -> “... is processed in a form...”

Page 3, line 58: “... feature pyramid and yolov3[17].” -> “... feature pyramid and YOLOv3 [17].” - The notation of “yolov3”, “YOLOv3”, “YoloV3” should be standardized in the whole paper.

Page 3, line 59: “... is introduced to reduces the required... and yields high accuracy.” -> “... is introduced to reduce the required... and yield high accuracy.”

Page 3, line 65-66: “... parts of aircraft... types of aircraft...” -> “... parts of an aircraft... types of an aircraft...”

Page 3, line 79: “... separable convolutions which are a form...convolutions. It factorize” -> “... separable convolutions are a form... convolutions. They factorize...”

Page 3, line 81: “... convolution apply a single filter...” -> “... convolution applies a single filter...”

Page 3, lines 82-85: “Compared to a standard convolution which input... one step. The depthwise...” -> “Compared to a standard convolution whose input... one step the depthwise...”

Page 3, line 86: “The convolution principle of depthwise separable and the standard is shown in figure3.” - There are grammatical errors, and therefore the meaning of this sentence is unclear.

Page 4, caption of Fig. 3: “Figure 3. On the left side of the equal sign are standard... pointwise convolution kernels.” -> “Figure 3. Standard convolution kernels... (left side) and pointwise convolution kernels (right side)”

Page 4, line 88: “... is the number of input channel...” -> “... is the number of input channels...”

Page 4, line 102: “Therefore, if we use 3 x 3 depthwise...” -> “Therefore, 3 x 3 depthwise...”

Page 4, line 106: “... for system predicts bounding boxes.” -> “... for predicting bounding boxes of the system.”

Page 4, line 111: “Take the feature map...” -> “Taking the feature map...”

Page 4, line 113: “... center of a object...” -> “... center of an object...”

Page 6, line 135: “... that generate three-layer...” -> “... that generates three-layer...”

Page 6, line 138: “... network output the 13 x 13 feature map...” -> “... network outputs the 13 x 13 feature map...”

Page 6, line 138: “Layer 11 concatenate...” -> “Layer 11 concatenates...”

Page 6, line 140: “... concatenate with...” -> “... concatenated with...”

Page 6, line 140: “The concatenate parts...” -> “The concatenated parts...”

Page 7, line 147: “... we regress bounding box...” -> “... we regress the bounding box...”

Page 7, line 152: “... confidence scores and are output by...” -> “... confidence scores are output by...”

Page 7, line 153: “... we only have crack class, the class probabilities is 1.” -> “... we only have a crack class, the class probability is 1.”

Page 7, line 154: “... of the predict bounding box... contains crack.” -> “... of the predicting bounding box... contains the crack.”

Page 7, line 161: “... data contains crack...” -> “... data containing the crack...”

Page 7, caption of Fig. 5: “Figure 5. The first line of four pictures are from different industrial equipments... interference. The second line of four pictures are from... with cracks.” -> “Figure 5. Different industrial equipments with cracks which have background interferences (a)-(d), aircraft structures with cracks (e)-(h)”

Page 8, line 190: “... and converge faster.” -> “... and can converge faster.”

Page 8, line 191: “... for our crack detection.” -> “... for crack detection.”

Page 8, line 202: “... the goal of network lightweight.” -> “... the goal of the lightweight network.”

Page 9, lines 208, 209: “... the network output the detection result.” -> “the network outputs the detection result.”

Page 10, line 241: “... standard convolution, which reduced...” -> “... standard convolution reduced...”

Page 11, line 247: “Future works include...” -> “Future works will include...”

Reviewer 2 Report

the paper presents a method for the detection of cracks in airplanes. A deep neural network is used to this purpose, adapting a method retrieved from the literature in order to decrease the time for training. A pyramidal structure is foreseen to detect cracks of three different scales. The proposed simplified structure of the network allows to reduce the calculation time, with a limited reduction of the performance.

Remarks

It is not clearly explained why the reduction of the training time is so strategic for this application. The inspections for the integrity of the fuselage usually are performed offline. Furthermore, the simplification allows to reduce the training time, which is performed offline. The authors should deepen this aspect.
